# COVID-19 Pandemic: Did Strict Mobility Restrictions Save Lives and Healthcare Costs in Maharashtra, India?

**DOI:** 10.3390/healthcare11142112

**Published:** 2023-07-24

**Authors:** Preshit Nemdas Ambade, Kednapa Thavorn, Smita Pakhale

**Affiliations:** 1Department of Population Health Sciences, Medical College of Georgia, Augusta University, Augusta, GA 30912, USA; 2Faculty of Medicine, School of Epidemiology, Public Health and Preventive Medicine, University of Ottawa, Ottawa, ON K1G 5Z3, Canada

**Keywords:** cost-effectiveness, COVID-19, coronavirus, India, Maharashtra, cost savings, lockdown

## Abstract

Introduction: Maharashtra, India, remained a hotspot during the COVID-19 pandemic. After the initial complete lockdown, the state slowly relaxed restrictions. We aim to estimate the lockdown’s impact on COVID-19 cases and associated healthcare costs. Methods: Using daily case data for 84 days (9 March–31 May 2020), we modeled the epidemic’s trajectory and predicted new cases for different phases of lockdown. We fitted log-linear models to estimate the growth rate, basic (R_0_), daily reproduction number (R_e_), and case doubling time. Based on pre-restriction and Phase 1 R_0_, we predicted new cases for the rest of the restriction phases, and we compared them with the actual number of cases during each phase. Furthermore, using the published and gray literature, we estimated the costs and savings of implementing these restrictions for the projected period, and we performed a sensitivity analysis. Results: The estimated median R_0_ during the different phases was 1.14 (95% CI: 0.85, 1.45) for pre-lockdown, 1.67 (95% CI: 1.50, 1.82) for phase 1 (strict mobility restrictions), 1.24 (95% CI: 1.12, 1.35) for phase 2 (extension of phase 1 with no restrictions on agricultural and essential services), 1.12 (95% CI: 1.01, 1.23) for phase 3 (extension of phase 2 with mobility relaxations in areas with few infections), and 1.05 (95% CI: 0.99, 1.123) for phase 4 (implementation of localized lockdowns in high-case-load areas with fewer restrictions on other areas), respectively. The corresponding doubling time rate for cases (in days) was 17.78 (95% CI: 5.61, −15.19), 3.87 (95% CI: 3.15, 5.00), 10.37 (95% CI: 7.10, 19.30), 20.31 (95% CI: 10.70, 212.50), and 45.56 (95% CI: 20.50, –204.52). For the projected period, the cases could have reached 631,819 without the lockdown, as the actual reported number of cases was 64,975. From a healthcare perspective, the estimated total value of averted cases was INR 194.73 billion (USD 2.60 billion), resulting in net cost savings of 84.05%. The Incremental Cost-Effectiveness Ratio (ICER) per Quality Adjusted Life Year (QALY) for implementing the lockdown, rather than observing the natural course of the pandemic, was INR 33,812.15 (USD 450.83). Conclusion: Maharashtra’s early public health response delayed the pandemic and averted new cases and deaths during the first wave of the pandemic. However, we recommend that such restrictions be carefully used while considering the local socio-economic realities in countries like India.

## 1. Introduction

In 2020, the world was devastated by a novel Coronavirus, SARS-CoV-2 (COVID-19). After first being reported in Wuhan, China, by the end of 2020, the virus had reached 219 countries [1]. Due to the novelty and high infectivity of the virus, many countries implemented non-pharmaceutical interventions (NPIs) such as quarantines and (partial or complete) lockdowns. The main intended benefit of lockdowns was the reduction of mortality and the avoidance of health system overloads [2]. Furthermore, they allowed governments sufficient time to build adequate healthcare infrastructure. Lockdowns restricted human mobility [3,4], and, thus, the spread of the virus. The global evidence suggests that strict mobility restrictions reduced COVID-19 cases and COVID-19-related deaths [5,6,7,8] in high-income [9,10,11] as well as low-and-middle-income countries (LMICs) [12]. The localized lockdowns were also impactful, especially during the resurgence of the pandemic after July 2020 [13]. Thus, from the perspective of avoiding mortalities and new cases, the literature suggests that there was a utility to the lockdowns.

However, the economic impact of lockdowns is unclear and has many caveats. For example, the benefits of lockdowns (in terms of reduced new cases and mortality rate) realized from reduced mobility are uneven across countries [14]. In the situation of waning population immunity and increasing threats from new deadly variants, the lockdowns could have been a cost-effective strategy [15,16]. In other instances, however, the overall cost–benefit impact of imposing a lockdown was not so promising. A study showed that the global burden of deaths due to COVID-19 was comparable with malaria, tuberculosis, and under-five mortalities due to pneumonia and diarrhea. It suggested that more devastation could perhaps be caused by prompting a recession [17]. Evidence also suggests that mitigation strategies comprising case isolations, the quarantining of house members, limited public gatherings, physical distancing, border controls, and restrictions targeting high-risk groups would generate more savings [2]. Furthermore, despite the proven cost-effectiveness of lockdowns, implementing them could be cost intensive [18,19]. Overall, lockdowns considerably prevented new cases and deaths; however, the cited studies suggested higher social and economic trade-offs for implementing the lockdowns.

The length and timing of implementing the lockdowns became critical considerations, and they are often ignored when measuring their cost effectiveness. Studies suggest that the optimal time period for implementing a lockdown could be 4 weeks [15], two months [12], or between 55 and 83 days [20]. Regarding the timing of implementation, researchers argued that the partial, inadequate, and/or delayed implementation of lockdowns was responsible for the continuous surge in COVID-19 cases [21,22]. Furthermore, over longer periods, lockdowns lose marginal benefits over other NPIs [15], they become economically unsustainable [16], and they carry risks of non-adherence [23]. Mobility restriction measures become ineffective as the pandemic progresses, and populations acquire immunity via immunization or acquired infection [24]. Therefore, a sound and well-followed epidemic plan is required to reduce the trade-off between benefits and societal and economic costs of NPIs.

India reported its first case of COVID-19 on 30 January 2020 [25]. Soon after reporting initial travel-related cases, concerned states implemented measures like contact tracing and quarantines. However, cases kept rising in the country, reaching a “local transmission” level by 16 March 2020 [26]. For the first time in its history, India announced a complete three-week lockdown at 8 PM on 23 March 2020, effective midnight 24 March 2020; it required 1.3 billion people to remain ‘home’. The lockdown’s first phase was implemented between 25 March 2020 and 14 April 2020 (21 days). During this phase, all industrial activities and public mobility were restricted. This phase was extended for an additional 19 days (Phase 2: 15 April 2020–3 May 2020), during which agricultural and essential services were allowed. The districts were divided into red (areas with 15 or more cases), orange (areas with less than 15 cases with no recent surge), and green (no cases) zones. Transport between the states facilitated migrant workers’ journeys to their home states. This phase was further extended for another 14 days (Phase 3: 4 May 2020–17 May 2020), with more relaxations in green zones, before being extended again for an additional 14 days (Phase 4: 18 May 2020–31 May 2020) owing to concern over rising cases. During phase 4, the states could demarcate containment zones within the red zones to implement stricter measures to allow local lockdowns within smaller geographical areas [27]. After the fourth phase, COVID-19 cases kept rising as the country gradually lifted restrictions. By 15 November 2021, India reported 34,456,401 COVID-19 cases—the second highest globally—and 463,852 deaths—the third highest globally [28]. India registered the highest daily death toll in the world during the second wave of the pandemic after June 2020 [29], as more contagious double mutant variants of the virus ravaged the country.

In India, the pandemic unfolded differently across the states [30,31]. For example, one study showed that the reproduction number significantly varied from 4.5 to 1.5 across the states during the different phases of lockdown [32]. According to another estimate, among the high-case incidence states, the reproduction number at the end of 15 and 30 days of lockdown varied between 1.72 to 5.72 and 1.37 to 3.99, respectively [33]. Salvatore and colleagues [30] estimated that the national-level reproduction number and case-doubling time by 31 May 2020 were 1.27 and 14.4 days, respectively. They reported the shortest (3.5 days) and longest (73.5 days) doubling rates in Assam and Punjab, respectively. By 14 July 2020, the doubling rate varied between 0.59 and 2.98 across all Indian states [34]. Among the COVID-19 cases reported from the country, the state of Maharashtra (MH) reported the highest numbers during the first wave of the pandemic (i.e., between March 2020 and May 2020) [29]. MH is one of India’s most industrialized states, constituting 9.42% of the country’s population [35]. Its contribution to India’s Gross Domestic Product (GDP) is 14%, which is the highest among all the states [36]. As of 31 May 2021, the state shared 20.3% of all COVID-19 cases in the country, which was the highest for any state [37]. Soon after the state reported its first case on 9 March 2020, the MH government constituted a “high-level empower action committee” on 13 March, issued its first COVID-19 guidelines for screening and home quarantine on 14 March, and closed public places on 15 March 2020 [38]. The state government also banned public gatherings and closed all non-essential services in affected metropolitan areas. Following the central government’s directions, the Maharashtrians observed a 14-h public curfew on 22 March 2020 [39]. The MH government also followed the federal government’s call for the lockdown [40]. The daily incidence of cases in the state for the entire lockdown period is shown in Figure 1, depicting a continued rise in new cases during this period. More than 3000 cases, the highest for the period, were reported on 23 May 2020. While the case numbers grew during this time, the mobility restrictions imposed by the lockdowns took effect. After 31 May 2020, the state implemented its “Mission Begin Again” plan to gradually open the economy [41]. However, the state remained a hotspot with the country’s highest number of cases, according to the reported data from India. By 15 November 2021, the state had recorded 6,624,986 cases and 140,602 deaths [42].

A notable amount of epidemiologic research related to the pandemic’s first wave [29] from India and MH state focuses on forecasting [43,44,45,46,47,48,49,50], explaining the disease’s clinical characteristics in certain populations [51,52], exploring environmental impacts [53,54,55,56], detailing adherence to protective measures [57], and assessing certain populations’ vulnerability [58,59,60,61]. Studies have projected the positive impact of severe mobility restrictions on the COVID-19 spread in the first wave [33,62,63,64,65,66,67]. Researchers have reported a significant reduction in human mobility [68,69,70] and anthropogenic heat [71,72] during the period the restrictions were in effect. Studies have also predicted a reduction in mortalities related to air pollution due to the mobility restrictions [73,74]. Importantly, the mobility restrictions also resulted in devastating impacts to migrant laborers when millions left their cities on foot and walked hundreds of kilometers to reach their rural homes [75,76,77]. Few studies have investigated the financial repercussions of imposing lockdown in the country. According to one estimate, the monthly economic loss during the lockdown in India was USD 64.1 billion, second behind the United States [9]. Barnett-Howell et al. [14] showed that given India’s (and other middle-and-low-income countries) younger demographic profile that was less susceptible to severe infection and mortality, fewer Quality Adjusted Life Years (QALYs) were saved with any public health intervention. Also, lockdowns resulted in fewer benefits in terms of costs, as higher economic costs were associated with them. The suboptimal health infrastructure also did not allow to gain any benefits from the lockdowns. As a result, the authors showed that the estimated benefits as a percentage of GDP from a 40-day lockdown (5.2%) was far lower than those achieved by implementing measures such as individual physical distancing (4.6%), closure of schools and workplaces (10.0%), and stay-at-home-orders (8.1%).

Bajiya and colleagues [78] focused on the post-lockdown period (1 September 2020 to 31 December 2020) to find a cost-effective strategy for reopening the economy. They stated that reducing transition rates and identifying infectious cases were key to flattening the curve. According to these researchers, efforts to implement a lockdown along with social distancing and enhanced awareness, testing diagnostics, and intensive medical care would prevent the highest number of cases. However, this approach would be the least preferable compared to the strategy in which lockdowns, social distancing, and awareness were implemented without providing intensive medical care. The Incremental Cost-Effectiveness Ratio (ICER) values for both strategies were 5.44 × 10^−7^ and 6.11 × 10^−8^, respectively, giving an edge to the latter over the former. The authors, however, did not clearly state how cost data were derived. Like Bajiya et al. [78], Mandal et al. [79] concentrated on finding a financially optimal strategy to implement while lifting the lockdowns. Using the city of Delhi as an example, they argued that an effective lockdown would cause a resurgence of cases and necessitate intensive testing and isolation to mitigate the effect. For effective reopening, they estimated that the weekly cost of testing would be between USD 26.68–42.12 million compared to the continued lockdown, which would cost between USD 1372–1979 million. As noted, the major caveat about both studies was that they focused on the post-lockdown period and did not provide estimates for the lockdown period.

Prinja et al. [80] provided cost-effectiveness estimates for the period of the lockdown. They used pre-lockdown (from 1 December 2019 to 22 March 2020) air travel data to model the disease progression during the 8-week-long lockdown. They found an 8-week lockdown could have shifted the epidemic’s peak by 34–76 days. The enhanced testing introduced after three weeks of lockdown could have helped prevent new infections and deaths by 70% and 60%, respectively, at the peak of the wave. Furthermore, they estimated that extending the lockdown beyond 8 weeks, along with the aggressive implementation of other NPIs, would have flattened the curve by the end of 20 weeks. Their cost exercise revealed that health system costs for managing the pandemic without a lockdown would have been USD 150 billion. Depending on the intensity of the public health measures, the total costs to the health system due to an 8-week lockdown would have been USD 340 billion. However, an extended stringent lockdown (at 80% effectiveness) beyond 8 weeks, along with other NPIs, would have cost only USD 0.03 billion to the health system. Thus, the authors’ economic analysis suggested a combination of extended lockdowns and NPIs as a cost-effective strategy to mitigate the spread. The study was one of the earliest from India to evaluate the impact of lockdown. Another strength of the study was that its modeling approach accommodated variation in intervention by examining the lockdown in phases. A major limitation of Prinja et al.’s [80] study was that due to differential population densities, mobility patterns, demographic composition, and the prevalence of comorbidities, the spread of the virus varied across the Indian states, which were not accommodated in the analysis. Prinja et al. [80], therefore, suggested carrying out a more granular state-level analysis to support local health policy decisions. Furthermore, the authors did not consider health system management and expansion-related costs that were required to manage the pandemic. Also, they could not study the negative impact of the pandemic on other healthcare needs, nor could they perform sector-wise macroeconomic analysis.

Despite observing inter-country variations in COVID-19′s spread, besides a few noted exceptions above, our knowledge about the pandemic is heavily based on what is reported from high-income countries [81]. Accordingly, our study addresses two major gaps in the previously mentioned literature. First, we aim to provide state-level estimates. For this study, we have analyzed the impact of harsh mobility restrictions during the first wave of the pandemic on disease spread and associated cost savings in the Maharashtra State—one of India’s COVID-19 reported hotspots during the first wave. Second, we incorporate health system expansion and program implementation costs to provide more realistic cost estimates. By conducting this analysis, we demonstrate a simple but useful way of modeling health systems costs that can be used to inform state-level health policies. Overall, our study provides a sub-national analysis of a low-and-middle-income country that could provide useful insights into the future pandemic containment strategies applicable in similar settings.

We used publicly available daily incidence data and available cost information from the literature (scientific articles and gray literature such as lay media and government reports) to provide informed estimates of COVID-19 transmission. We compared the cost of averting new COVID-19 cases by implementing strict mobility restrictions against allowing unrestricted spread of the infection.

## 2. Data

The study’s total duration was 84 days, from 9 March 2020 (the day on which the first COVID-19 case was reported in Maharashtra) to 31 May 2020 (the last day of phase 4 of lockdown). Data from the period from 14 March 2020 to 31 May 2020 were collected from a crowdfunded, publicly available data source: COVID-19 INDIA TRACKER (https://www.COVID19india.org (accessed on 25 June 2020)) [82], while the data from 9 to 13 March was collected from a report published on 1 April 2020 by Maharashtra’s Medical Education & Drugs Department (MEDD) [83].

## 3. Analysis

### 3.1. Epidemic Trajectory Analysis

In the first section, we mapped the trajectory of the COVID-19 pandemic in Maharashtra during the lockdown using log-linear models fitted on a time series of daily incidence cases. The first section is influenced by the analysis presented by Tim Churches [84,85]. We mainly used analytical packages developed by the R Epidemics Consortium (RECON) for the epidemic trajectory and projection analysis. The “incidence” package [86] was used to fit the following log-linear model on daily incidence data:Log (y) r × t + b(1)
where ‘y’ is the incidence, ‘t’ is time (in days), ‘r’ is the growth rate, and ‘b’ is the origin. We estimated separate models for each phase of lockdown, considering the first day of the phase as the origin.

Transformation of the growth rate into the reproduction number (R_0_) is possible if the serial interval distribution is known. Following the approach of Wallinga and Lipsitch [87] implemented in the “epitrix” package [88], we used the previously estimated growth rate for each lockdown phase to derive 1000 samples of the R_0_. We then calculated the mean R_0_ from the derived samples by using the mean serial interval (SI) of 3.9 days and the mean standard deviation (SD) of 2.85 days reported for India [89]. 

We also estimated the time-varying reproduction number (i.e., instantaneous effective reproduction number: R_e_) given that the mobility restrictions changed during the different phases of lockdown, employing the method of time-varying R_e_ as described by Cori et al. [90] and later extended by Thompson et al. [91] Considering the rapid spread of infection in Maharashtra, we chose a smaller moving window of five days to explicate the time trend shown by R_e_. The mean (minimum 2.97 days, max. 7.5 days) and SD (minimum 2 days, max. 10.9 days) of SI estimation for COVID-19 vary across the literature [89,92,93]. To accommodate this uncertainty around the SI estimation, we drew 1000 pairs of mean and SD of SI ranging between their reported maximum and minimum values and then drew a posterior sample n = 1000 for each pair. The R_e_ was then estimated using this posterior distribution of the mean, SD of SI. We performed R_e_ estimation analysis using the “EpiEstim” package [94]. Researchers have widely used this package to determine the epidemic trajectory in different settings [34,91,95].

After estimating R_e,_ we measured the impact of lockdown by comparing the cumulative incidence of cases in the scenario in which the infection spread without any restrictions against the lockdown measures. Our preliminary analysis showed the delayed effect of mobility restrictions (denoted as lockdown in the figures) on the disease spread. Therefore, we estimated initial R_0_ based on combined daily case incidence data for the pre-lockdown and lockdown-1 period. Based on this R_0_, we made projections for the remaining phases (i.e., lockdown phase 2, 3 & 4) using the “projections” package [96] (scenario A) and compared it with actual cases (scenario B) reported for the same period. The forecasting was compared with the pre-and lockdown-1 period’s actual data to check the projections’ acceptability. We used “ggplot2” [97] for data visualization purposes. All packages above were implemented in R statistical software version 4.0.3 [98].

### 3.2. Economic Impact Analysis

We compared costs for the unrestricted spread of COVID-19 against the imposed mobility restrictions from the health system’s point of view. As stated before, the projected cases for the second, third, and fourth lockdown periods were the number of cases if the virus was allowed to spread without any intervention during the given period. The direct cost estimation is based on the following variables derived from the projected cases using the following rates reported in the literature: hospitalization rates, intensive care unit (ICU) admission rates, death rates, and average length of stay (LOS) in the hospital and the ICU. Based on these parameters, we calculated the direct health system costs for the scenarios by adding the equations given below
no. of individuals in ICU care * [(cost of PPE + cost of medicine + cost of ICU care) * LOSIa (7 days) + cost of test kit] (2a) (ICU patients)no. of individuals in ICU care * [(cost of PPE + cost of medicine + cost of general care) * LOSIb (7 days)] (2b) (post-ICU care in general ward for ICU patients)no. of individuals hospitalized in non-ICU care * [(general ward cost + cost of PPE + cost of medicine) * LOSH (14 days) + cost of test kit] (2c) (non-ICU patients)no. of non-hospitalized cases in isolation * (cost of test kit + cost of outpatient medicine) (2d) (non-hospitalized isolated patients)(2)

Details about the assumptions, unit costs, and associated references for direct and indirect cost estimates are provided in Table 1. For our analysis, we assumed 15 percent of reported cases would require hospitalization. Furthermore, based on the state public health department’s reports, we assumed two percent of reported cases would require critical care in the ICU. The case fatality rate reported at the end of the first lockdown was four percent for the state. The average length of stay for ICU and non-ICU patients being treated for COVID-19 was 7 (range 4–19 days) and 5 (range 4–21 days) days, respectively. However, for the first few days, most of the cases were kept under mandatory 14-day care in the state. Therefore, for our base case, we assumed all hospitalized individuals (ICU and non-ICU alike) spent two weeks in the hospital. Thus, the patients receiving 7 days of ICU care spent an additional 7 days in general care, while the non-ICU patients spent 14 days in general care. In our sensitivity analysis, we varied the general care stay for ICU patients and other length of stay parameters (described below). Presuming there is no shortage, up to 15 PPE kits can be used daily to treat one COVID-19 case, as per the European CDC estimates. Additionally, we anticipated that each identified case would utilize three COVID-19 test kits. In the first wave of the pandemic, there were no reliable estimates for hospitalization costs available for COVID-19 related admissions. However, the MH government later capped private hospital charges at INR 4000 ≈ USD 53.33 (general ward with isolation), INR 7500 ≈ USD 100 (ICU care without ventilator support), and INR 9000 ≈ USD 120 (ICU care with ventilator support) Indian rupees. These rates excluded PPE kits, high-end medicines, and testing charges. Due to the lack of exact cost data on ICU expenditures for COVID-19 patients, we used daily expenditure estimates of INR 8825 (≈USD 118) for a Chronic Obstructive Pulmonary Disease (COPD) patient admitted in an ICU. For non-ICU hospitalizations, we considered INR 4000 as daily care charges. Based on the media reports, our calculated 14-day cost for PPE, medicines, and test kits is INR 185,000 (≈USD 2467). Using previously mentioned assumptions and estimates, the 14-day per-patient direct cost for individuals in ICU and non-ICU care totaled INR 274,775 (≈USD 3664) and INR 241,000 (≈USD 3213), respectively. We assumed the non-hospitalized cases were kept in 14-day institutional isolation and incurred testing and outpatient medicine charges (INR 500/USD 7).

For the indirect costs, as shown in Equation (3), we calculated the loss of income for the total Length of Stay (LOS) in cases of hospitalization/quarantine/isolation and total loss years due to premature death, representing the opportunity cost of the infection. A premature, COVID-19-related death is a death occurring at an age younger than the country’s average life expectancy.
Indirect cost = [no. of individuals in ICU care * [(LOSIa (7 days) + LOSIb (7 days)] + no. of individuals hospitalized in non-ICU care * LOSH (14 days)] * loss of daily wage + no. of cases in isolation * LOSQ (14 days) * loss of daily wage + no. of home/institutional quarantine contacts * LOSQ (14 days) * loss of daily wage + no. of premature deaths * average loss of years * loss of yearly wage.(3)

We have included wage losses for proximate contacts in our indirect costs estimates, as shown in Equation (3). It is important to isolate or quarantine proximate contacts of identified cases to prevent the infection’s secondary spread. Unfortunately, studies reporting the average number of people coming into contact with an infected COVID-19 case from the MH is not available. Evidence from rural parts of the neighboring state of Gujarat suggests that a COVID-19 case makes contact with an average of 5.3 people during their entire infection period [111]. However, given the high urban population density and connectivity in Maharashtra, this number could be higher. Based on our informal communications with state and district public health officials, we assumed each COVID-19 case encounters 20 proximate contacts, which will require home or institutional quarantine. Accordingly, we derived the total number of proximate contacts and associated opportunity costs of home/institutional quarantine in each scenario.

The indirect costs included the opportunity cost of contracting the COVID-19 virus and being exposed to the identified case. We considered 14 days as the average number of days lost due to COVID-19-related hospitalization, isolation, and quarantine. Based on the COVID-19 data reported until 15 April 2020 [100], our calculations (Appendix A) showed that the average age of death was 58 yrs., and nearly 84% died younger than the country’s overall life expectancy of 68.3 yrs. [104]. Thus, the average loss of life years in case of a premature COVID-19 death was 10.3 life years. The 14-day and yearly loss of income calculations are based on the expected average per capita income of INR 191,827 (≈USD 2558) estimated for Maharashtra in 2018–19. We calculated an average daily wage of ≈ΙΝΡ 761/USD 10 based on the total number of working days: 252 for the 2018–2019 financial year. As explained in the methods section, we followed the 1:20 (case:contact) ratio to estimate the probable number of high-and-low-risk contacts made by each case, which were further put under either home or institutional quarantine for 14 days (see Appendix A for data). Given the relatively younger population of the state and country, we assumed all these contacts belonged to the working-age category and had to face opportunity costs for quarantining. The wage loss due to hospitalization, death, and quarantine or isolation for both scenarios was calculated accordingly.

The aggregate cost of COVID-19 infection for each scenario is the sum of Equations (2) and (3). We considered funds allocated by central and state governments for controlling the COVID-19 pandemic as the fixed costs for tackling the infection in both scenarios. We assumed in both scenarios that the costs of COVID-19 medical treatment were paid through National Health Mission’s (NHM) budget, as suggested in the federal government’s communication to the states [112]. The Maharashtrian government initially allocated INR 450 million (≈USD 6 million) to tackle the COVID-19 pandemic [113]. Based on information provided in parliament by the Minister of State in the Ministry of Health and Family Welfare [114], we calculated that the state received a total of INR 7.26 billion (≈USD 96.85 million) in the form of grant-in-aid and equipment under India’s COVID-19 Emergency Response and Health System Preparedness Package disbursed as of 10 September 2020 (see Appendix A). Furthermore, 35% of the State Disaster Relief Management Fund (SDRMF) yearly allocation (i.e., INR 7.52 billion ≈ USD 100.24 million) was allowed for COVID-19-related expenditure [112,115]. Along with these funds, as per the news reports, the state government utilized INR 238.2 million (≈USD 3.18 million) from its Chief Minister’s COVID-19 Relief Fund for COVID-19-related medical care during the lockdown period [116]. Thus, roughly INR 15.47 billion (≈USD 206 million) was allocated/expensed to tackle the pandemic for the given period.

The difference between the total costs for both the scenarios gives the value of total averted infections is shown below
Value of Averted Infections = Total costs (scenario A) − Total costs (scenario B)(4)

The total savings from the averted cases were calculated as a percentage value of averted infections based on the total costs for scenario A.

We have used media sources, reports, guidelines, government documents, and published literature to calculate costs and probabilities and have created an approximate estimation whenever such information was unavailable. All costs were calculated in Indian rupees (INR) and converted to US dollars, assuming INR 75 = USD 1. Furthermore, we conducted a sensitivity analysis to accommodate the variability in our input cost variables. The input parameters (Appendix A) included in the analysis were: length of stay in ICU care (4–19 days), days spent in general care post ICU treatment (3–10 days), length of stay for non-ICU hospitalizations (4–21 days), total quarantine days (7–21 days), ICU bed charges (INR 7500–9000), and PPE kit cost (INR 500–1000).

## 4. Patient and Public Involvement

This research does not directly include patient or public involvement. The project objective and research question are informed by the national and international experiences of the authors during the ongoing pandemic.

## 5. Results

### Epidemic Trajectory Analysis

Our estimates for case doubling days and reproduction numbers presented in Table 2 show that these restrictions have delayed the spread of the infection to the larger population. Figure 2 depicts the visual representation of the log-linear models for different phases of the lockdown. The case-doubling rate decreased to a mere four days in phase 1 from 17 days in pre-lockdown, indicating the beginning of the rapid spread of COVID-19 in MH. During the later stages of lockdown, this rate reached ten days (phase 2), 20 days (phase 3), and 45 days (phase 4). The estimated reproduction number (R_0_) varied significantly across the different phases of the lockdown. As shown in Table 2, the pre-lockdown R_0_ was 1.14, which increased to 1.67 during the lockdown-1 period. The effects of the lockdown lagged behind the actual implementation, as the estimated R_0_ came down to 1.24 in lockdown phase 2 and 1.12 in lockdown phase 3, respectively. It reached 1.05 in the last phase of the lockdown.

The instantaneous R_e_ as shown in Figure 2 is high for the pandemic’s early days, even after the initiation of the first lockdown phase, with a downward trend for the later period.

Based on the estimated median basic reproduction number of 1.473 (95% CI: 1.39, 1.56) for the combined pre-and-lockdown-1 period, the cumulative number of projected cases for the remaining period of lockdown (i.e., phases 2, 3, and 4) was 631,891. The actual number of cases reported for the same period was 64,975. Figure 3 is the visual representation of the daily observed COVID-19 cases versus projected COVID-19 cases. According to our model, without the mobility restrictions, the number of infected cases in MH could have reached more than 60,000 cases per day by the end of May 2020. However, the mobility restrictions helped check the spread; thus, the daily case tally reached 2487 cases on 31 May 2020. We compared the projected cases from the pre- and lockdown-1 period with the actual cases reported for the same period to check the validity of our projection. The results (not presented here; see Appendix A) showed that our projections aligned with the actual reported cases.

## 6. Economic Impact Analysis

Table 3 elucidates the economic impact of COVID-19 cases with and without mobility restrictions. We compared the cost of allowing the unrestricted spread of COVID-19 in Maharashtra against the imposed lockdown measures. Following Equation (2), the total direct costs for projected and actual cases during the lockdown 2, 3, and 4 periods were INR 33,304.13 million (≈USD 444 million) and INR 3460.93 million (≈USD 46 million), respectively. The total indirect costs corresponding to scenarios A and B were INR 182.91 billion (≈USD 2.44 billion) and INR 18.02 billion (≈USD 240 million), respectively. The estimated total costs for scenarios A and B were 231.69 and 36.96 billion Indian rupees (≈USD 3.09 and 0.49 billion), respectively. Based on Equation (4), the difference between total costs gives the total value of averted cases: INR 194.73 billion (USD 2.60 billion). Thus, the total cost savings realized by imposing strict mobility restrictions for the given period was 84.05 percent.

Next, we discuss our results in the context of the state’s GDP and calculate ICERs (The GDP and ICER-related analysis and results are added following the reviewers’ suggestions). In 2019–2020, the year before the pandemic, MH’s GDP in real terms was INR 21,544,460 million (USD 287,259 million) and its population was 112.37 million [117]. Our calculated total cost for the unrestricted spread of the virus was 1.08% of the GDP. The total value of the lockdown was 0.17%, and the total value of averted cases was 0.90% of the GDP. Furthermore, we estimated ICER for the total number of estimated deaths using the GDP-based approach without incorporating discounts in health and cost values. To this end, we derived per-case costs for both interventions by dividing the total costs by the total number of cases for each approach (please see Appendix A). The estimated costs per case for the projected and lockdown scenarios were INR 366,654.56 (USD 4888.73) and INR 568,764.10 (USD 7583.52), respectively. Next, we used these unit costs to derive total mortality costs by multiplying them by the total number of deaths in each scenario. The total mortality costs calculated with this approach were INR 9267.56 million (USD 123.57 million) and INR 1198.95 million (USD 15.99 million) for projected and lockdown scenarios, respectively. Using total mortality costs and deaths from both strategies, the incremental costs and deaths were INR 8068.61 million (USD 107.58 million) and 23,168 deaths. The resulting ICER (ratio of incremental costs and incremental death) was INR 348,265.11 (USD 4643.53), which reflects the cost of preventing one death. Furthermore, we considered the average number of years lost (10.3 years) as the total QALY lost and assumed that if one death was equivalent to a loss of 10.3 QALYs, then ICER per QALY was INR 33,812.15 (USD 450.83). Using the 2019–2020 value of GDP, the per capita GDP in real terms was INR 191,721.04 (USD 2556.28). According to the WHO-CHOosing Interventions that are Cost-Effective (WHO-CHOICE) threshold, the ICER should be less than three times the per capita GDP when considering cost effectiveness [118]. Using this criterion, our calculations suggested that imposing a lockdown was a cost-effective intervention. This conclusion was valid even if we use another threshold of USD 487 suggested for India [119].

## 7. Sensitivity Analysis

The results show that the total number of quarantine days has the largest effect on total cost savings, followed by non-ICU hospitalization days and costs of PPE kits. However, in all scenarios, imposing harsh mobility restrictions remained the dominant cost-saving strategy. Our estimates show minimal sensitivity for the rest of the parameters in the analysis. The related table and tornado diagram are presented in Appendix A.

## 8. Discussion

We mapped the epidemiologic trajectory of the COVID-19 pandemic in its initial days for the state of Maharashtra in India. Based on both case-doubling rates and basic reproduction number analysis, we can infer that imposing strict mobility restrictions delayed the rapid spread of COVID-19 in MH. Such delays allowed the governments to buy more time and prepare the health infrastructure for future phases of the pandemic. It is clear from Table 3 that these restrictions averted a larger number of COVID-19 cases and saved medical costs for the state. Our estimate of total costs in case of unrestricted spread (1.08% of the state GDP) aligns with Prinja et al’.s [80] estimate of 4.5% at the national level. The intended benefits of the lockdown at the national level are reported to be 5.2% of the GDP [14], whereas we report they could be almost one percent of MH’s GDP. By comparing national-level costs and savings with our state-level estimates, the unrestricted spread of the virus in MH could have had a devastating impact on the country’s economy. The savings accrued due to averted cases in the state also significantly contribute to total benefits at the country’s level. These savings are important given the state’s population size and contribution to India’s economy. However, compared with the state’s GDP, our estimates were relatively smaller than anticipated. This might be because our calculations only accounted for costs and savings to the health system and excluded other societal costs and savings. Thus, the costs and savings of managing the pandemic with and without the lockdown could be much higher than we have estimated.

It is also important to note several factors that could produce different results in economic evaluations similar to ours. One such factor is the variability [19], as there are pros and cons [120] associated with using currently accepted thresholds for cost-effectiveness benchmarking. Another factor is considering the short or long-term impact of the lockdown. The economic benefits and losses from the lockdown could be spread over a long period, and its implications might reveal different results. For example, Lally [2] has accounted for more than a year’s worth of data from Australia and has shown that locking down populations was not an economical strategy compared to a milder approach comprising testing, isolation, and quarantining. However, as discussed in the introduction, the evidence is still inconclusive. After reviewing the literature, we infer that the results depend on the data, the time period considered, the threshold used, and the approach adopted, which is true for all economic evaluations. Nevertheless, as Bajiya et al. [78] have shown, the lockdown’s prolonged implementation is not recommended and might not prove cost effective. Testing and serological surveillance [79], and wearing masks along with quarantining and isolation [12], have proven beneficial when reopening after a lockdown.

Even though our estimates are conservative, we have demonstrated an approach to compare NPIs at the state level by including a detailed cost profile of the parameters. Our study complements international [14] and national [78,79,80] studies and fills an important gap in the literature by performing a more granular analysis. To the best of our knowledge, this is the first study to assess the financial implications of the mobility restrictions imposed in MH based on real-time data. This study will help create evidence-based health policies for the state in the ongoing and future management of pandemics. Despite high anticipated societal costs, governments across the world implemented lockdowns. Agreeing with our calculations, studies (as discussed in the introduction) concluded that lockdowns successfully halted the pandemic and delayed the peak of the cases. Research shows that countries with better governance quality could attain a higher reduction in new COVID-19 cases during the lockdown than countries with relatively lower governance quality [121]. Lally [2] suggested several reasons (1) fearing worse outcomes because of a lack of historical experience; (2) the need for swift action; (3) fear of being blamed for higher immediate deaths due to non-mitigation; (4) the potential of costs being distributed over a longer period; and (5) lack of emotional appeal of the GDP loss that might have motivated policymakers to implement lockdowns in early phases of the pandemic.

Some researchers have questioned the usability of lockdown as an NPI to stall the pandemic. Tanovskiy and Socol [122] discussed the pre-pandemic evidence to confirm the absence of lockdown as an NPI. They further pointed out the arbitrariness of the cost-effectiveness threshold and demonstrated that the QALYs saved for COVID-19 patients were generally similar to those of other chronic diseases. Conversely, other researchers are certain about the lockdown’s impact on community mobility. Skepticism arises because lockdown has not proven to be the optimal solution in many settings (see the discussion in the introduction). According to Lally [2] and Spiliopoulos [4], the reasons behind this phenomenon are (1) uncertainty in determining the directionality of the cause–effect relationship between death rates and policy decisions (reverse causality); (2) stringent mobility restrictions initiating a “signaling effect” [4] among people prompting a voluntary change of behavior; (3) increased chances of high-risk transmission (e.g., within home transmission) offsetting the effect of population-level reduced mobility; (4) late initiation; and (5) limited time-series data or lack of explicit behavioral modeling. We want to note that the lockdown’s success depends on how quickly the health systems are strengthened before relaxing the restrictions. For a low-resource setting such as India, symptom-based household quarantining and compulsory mask-wearing are the suggested strategies during the reopening period [12].

Besides discussing the lockdown’s epidemiologic usability, it is imperative to mention its unintended consequences. On a positive note, fewer traffic injuries and fatalities occurred during lockdowns, and a drop in air pollution, crime, and overall mortality was observed [15]. However, the human costs of lockdowns are noteworthy [122]. Stricter mobility restrictions have widened socioeconomic and health disparities. The pandemic has increased the vulnerability of migrants in India [77] and internationally [123]. The lockdowns have negatively affected physical, social, and mental health [15,17,77]. They have altered the workplace atmosphere, disrupted children’s education [15,17,77], and fractured social relationships [124]. These unintended negative consequences challenge the implementation of lockdowns as an effective NPI.

Weak public health infrastructure along with high population density; high risks of contracting infection due to occupational exposure; high proportions of migratory populations; poverty; and congested housing make restricting human mobility for disease containment a challenging task in LMICs like India. Despite saving on medical costs, as this paper has demonstrated, abruptly implementing harsh mobility restrictions resulted in the concomitant loss of lives and livelihoods, especially for urban migrant workers. With the sudden announcement of the lockdowns, nearly 11.4 million migrant workers haphazardly left cities for their homes due to a loss of shelter and livelihood; at least 971 died due to this plight [125]. Moving forward, Marcu [123] and Panneer et al. [77] demand concrete and safe policies on employment, livelihood, food, and social justice for migrants. They further recommend a stronger support system in the home states of migrants.

As with other countries, international travelers brought COVID-19 to India [126]. The wealthy and influential among these travelers sidestepped public health guidelines [127]. Perhaps acting immediately on international warnings and isolating inbound international travelers could have saved more lives and costs to the country and could have protected the most vulnerable citizens working in the informal economy and living in crowded settings. Moreover, the experience of other Asian countries such as China, Singapore, South Korea, and Taiwan has shown that aggressive testing, contact tracing, and isolating cases, especially strictly applying these strategies to international travelers, could have helped successfully mitigate coronavirus and flatten the curve [128,129,130].

It is worth noting the limitations of our study before we conclude. To begin, our model is simple and does not incorporate delays in case reporting could be a limitation. A more nuanced study utilizing the data between January 2020 and December 2021 (the first and second wave of the pandemic) has estimated that the number of deaths that occurred during this period was almost four times higher than that in MH’s reported data [131]. However, by focusing on the early pandemic, our results are aligned with other studies showing that mobility restrictions effectively reduced cases in the country [30,32,33,44,45,46,64,66]. Our assumption that the pre-and-lockdown-1 period reflects infectivity in the conditions in which the spread of the virus was allowed without restriction is supported by the instantaneous R_e_ results. We report these values reaching up to 3.5 in the initial days of the pandemic. This could be due to the shorter serial interval than the incubation period or the higher presence of undetected asymptomatic, sub-clinical spreaders of COVID-19. We have accounted for the uncertainty around serial interval estimates in our analysis to minimize the chances of unreasonably high reproduction numbers. The literature and media reports suggest that the number of asymptomatic patients was higher in India [132], and per capita testing performed in the first few days of the pandemic was inadequate [133]. Therefore, many asymptomatic individuals might have gone undetected during this period, spreading the infection to their close contacts even during the lockdown. Also, during the same time, an exodus of migrant workers flowed towards the rural parts of the state and other parts of the country, further spreading the infection to these areas. These events might have caused a continued rise in cases despite the mobility restrictions.

Another limitation is estimating the number of contacts based on disease infectivity. International studies suggest that connectedness matters more than population density when explaining the spread of COVID-19 in denser areas [134,135]. However, India-based studies have reported a positive but moderate correlation between population density and COVID-19 cases [136,137,138]. In Maharashtra’s case, its well-connected and densely populated urban centers saw much greater growth of COVID-19 cases than the distant rural areas. Accordingly, the number of traceable, exposed contacts varied across the districts, ranging from 27 in Palghar to only 3 in the Nandurbar district [139]. For scenario B in particular, we could have based our contact estimation on media statements published daily by the state public health department. However, only cumulative numbers of institutionally or home- quarantined individuals were mentioned in these reports, from which those who completed the quarantine or were newly quarantined could not be estimated. Therefore, we estimated the total exposed contacts based on the average state contact tracing figures that officials reported. One point to note is that in the initial days, the state’s average traceable contacts were low and fluctuated even within metropolitan areas such as Mumbai [140,141,142,143]. The Indian Council of Medical Research (ICMR)—the country’s apex organization for biomedical research—issued guidelines on contact tracing to target 20 contacts per case [144]. Our estimates of the number of contacts matched the probable exposed contacts per case that would have required isolation/quarantine to curtail secondary transmission.

The unavailability of data also constrained the direct cost estimations for hospitalization for those who died in the hospital. We assumed similar ICU and non-ICU medicine costs, which might have resulted in underestimating direct costs. As our analytical time window was small, we did not account for long-term benefits gained by purchasing equipment (e.g., ventilators) while calculating government expenditure. Also, we did not have data for the occupations and age ranges of the quarantined contacts; hence, precise estimates of wage losses could not be made. We assumed everyone in quarantine belonged to the working-age group and estimated wage losses accordingly, which might have resulted in overestimation. Moreover, we have not considered the lockdown phase 1 period for cost analysis because its effect on new cases was realized in later phases. Furthermore, the loss of wages for inbound and outbound migrant workers who were temporarily quarantined or for those who lost their jobs during the period because of the shutdown of their employment activities (e.g., factory workers, daily wage laborers) was not considered for calculations. Therefore, the actual economic impact of the lockdown might be larger than our estimates, given these limitations and our conservative approach to cost estimations and approximations.

Finally, though not captured in this analysis, the psychological costs of both the COVID-19 infection and the resulting lockdowns were also huge. Minority communities and healthcare workers were constantly stigmatized and ostracized during the pandemic [145]. Likewise, the costs of disruptions to social relationships caused by lockdowns are also not captured in the literature. However, measuring these indirect costs will require detailed data on economic losses from different sectors of the economy. It will take several years to realize the full impact of this ongoing pandemic. Future studies should evaluate the full economic implications and costs of the pandemic’s hefty psychological and social tolls.

## 9. Conclusions

We conducted a state-specific economic analysis based on data from Maharashtra, India and have shown that strict mobility restrictions, despite their upfront costs, delayed the spread of disease and ultimately saved money for the healthcare system. Our analysis shows a simple approach to conducting economic analysis at the subnational level that can inform local public health policymaking. However, despite the savings for the health system, strict lockdown measures can also devastate people’s lives and livelihoods unless backed up by strong social programs. Therefore, we recommend that such restrictions be carefully used while keeping in mind the local socio-economic realities in countries like India.

## Figures and Tables

**Figure 1 healthcare-11-02112-f001:**
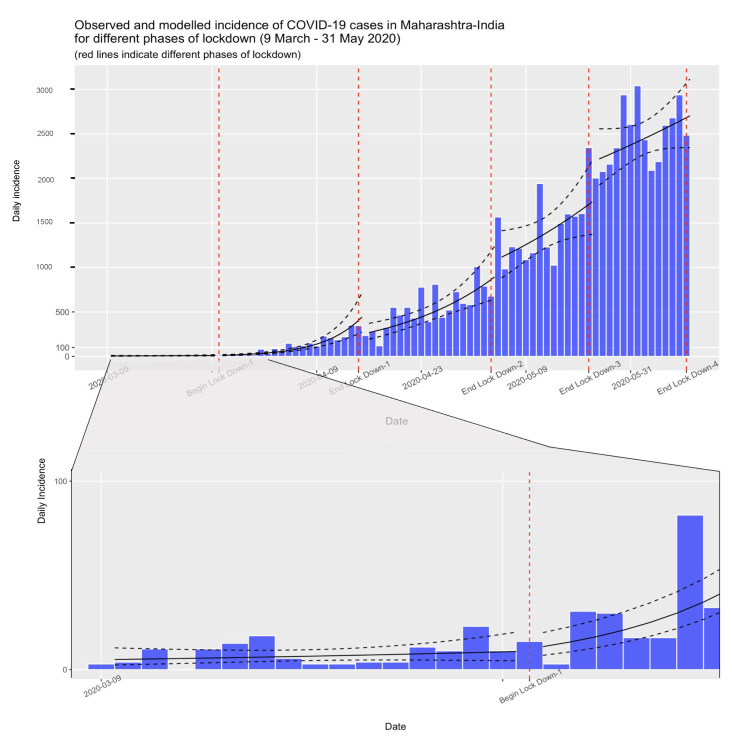
Observed and modeled (shown as a solid black line and 95% Confidence Intervals as dotted lines) incidence of COVID-19 cases in Maharashtra, India (9 March–31 May 2020).

**Figure 2 healthcare-11-02112-f002:**
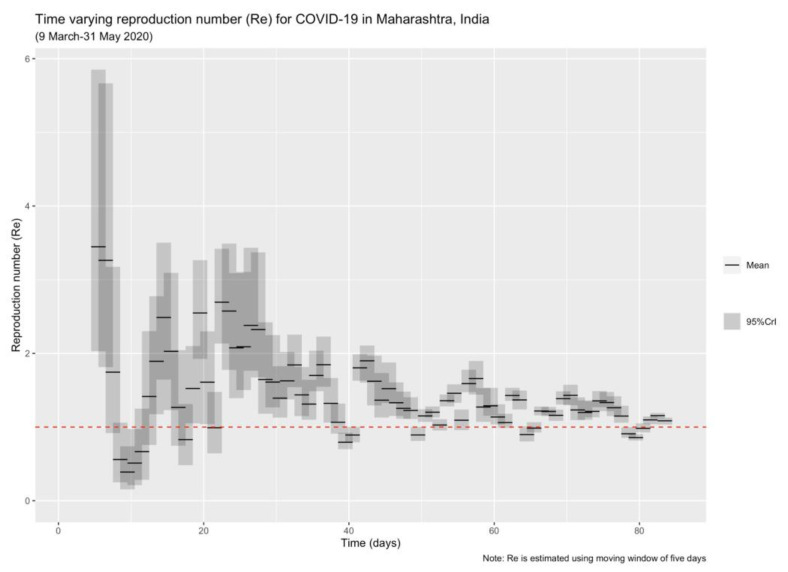
Time–varying reproduction number (R_e_) for COVID-19 in Maharashtra, India (9 March–31 May 2020). The red dashed line indicates threshold R_e_ = 1 when, on average, each infected individual infects only one other individual.

**Figure 3 healthcare-11-02112-f003:**
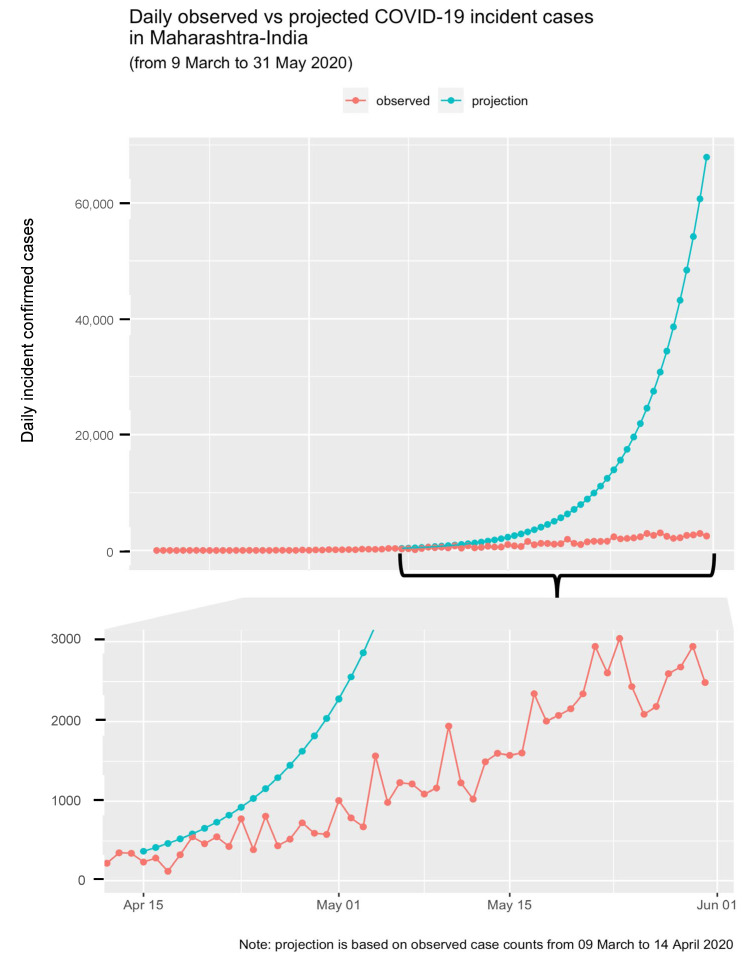
Daily observed vs. projected COVID-19 incident cases in Maharashtra, India.

**Table 1 healthcare-11-02112-t001:** Input parameters, assumptions, and associated costs.

Input Parameters	Unit Values	Calculation Notes
Probability of hospitalization after COVID-19 infection [99]	0.15	
Probability of ICU care after COVID-19 infection [100,101]	0.02	Based on MEDD reports published on 15 April 2020 and 12 May 2020.
Number of deaths due to COVID-19/probability of death [100]	0.04	Based on MEDD Report 15 April 2020 fig16
Average length of stay for non-ICU hospitalization (Sensitivity analysis range 4–21 days) [102]	14 days	Authors’ assumption based on treatment and quarantine guidelines of that time. Range 4–21 days used for sensitivity analysis.
Average length of stay for ICU [102] (range 4–19 days)	7 days *	A range of 4–19 days was used in sensitivity analysis.
Average length of stay for ICU patient in general ward post ICU care (range 3–10 days)	7 days *	Authors’ assumption.
Income per capita/year for Maharashtra state [103]	191,827 (INR)	Estimated annual per capita income as per the report for the year 2020.
Average daily wage	761.218254 (INR)	Working days in 2018–2019 from 1 April to 31 March = 252 days used to calculate daily per capita wage from annual estimates.
Average life expectancy [104]	68.3 yrs.	
Average age of dying due to COVID19	58 yrs.	Authors’ calculations based on table-4 of MEDD report published on 15 April 2020.
Number of years of potential life lost	10.3 yrs.	Subtracting average age of dying from average life expectancy
Average days of quarantine [105]	14 days	A range of 7–21 days used for sensitivity analysis.
Number of PPE required per patient per day [106]	15	As per estimates from European CDC
Cost of PPE (range 750–1000 INR) [107]	750 (INR)	A range of INR 500–1000 used for sensitivity analysis
Cost of Inpatient medicine (daily) [107]	1000 (INR)	
Cost of Outpatient medicine (for entire period)	500 (INR)	Authors’ assumption
Cost of test kit/test [107]	4500 (INR)	
Cost of ICU care (per day) [108,109,110] (range 7500–9000 INR)	8825 (INR)	This figure is based on per day ICU stay cost including ventilator support for COPD patient. For sensitivity analysis range is used.
Non-ICU General ward bed charges [109,110]	4000 (INR)	
Loss of income (for 14 day)	10,657 (INR)	Average daily wage X Average days of quarantine
Loss of income due to premature death	1,975,818.1 (INR)	Number of years of potential life loss X Income per capita per year for Maharashtra state

* Note: Beyond the ICU stay, the patient was assumed to remain in hospital for 7 days in general care before the discharge.

**Table 2 healthcare-11-02112-t002:** Median reproduction number (R_e_), Case growth rate, and case doubling rates for different phases of lockdown based on log-linear model fit for COVID-19 cases in Maharashtra, India (9 March–31 May 2020).

Phase	Case Doubling Rate in Days ^#^	Median Reproduction Number (R_0_) ^#^
Pre-lockdown (9–24 March)	17.78 (5.61, −15.19)	1.14 (0.85, 1.45)
Lockdown-1 (25 March–14 April)	3.87 (3.15, 5.00)	1.67 (1.50, 1.82)
Lockdown-2 (15 April–3 May)	10.37 (7.10, 19.30)	1.24 (1.12, 1.35)
Lockdown-3 (4–17 May)	20.31 (10.70, 212.50)	1.12 (1.01, 1.23)
Lockdown-4 (18–31 May)	45.56 (20.50, −204.52)	1.05 (0.99, 1.12)

Note: Median reproduction number estimates are bootstrapped over 1000 iterations and calculated by assuming mean serial interval and standard deviation of 3.9 and 2.85 days, respectively. ^#^ 95% Confidence intervals are in parenthesis.

**Table 3 healthcare-11-02112-t003:** COVID-19 infection spread scenarios and associated costs.

Case Categories and Estimated Costs	Projected	Actual
Total cases for the projected period	631,891	64,975
Hospitalized individuals	94,784	9746
Individuals admitted to ICU	12,638	1300
Total number of deaths	25,276	2108
Isolated infected individuals	499,194	53,929
Number of premature deaths	21,138	1763
Total contacts quarantine/isolated	12,637,820	1,299,500
Direct Cost due to hospitalization *		
ICU patient cost	3472.56	357.07
Non-ICU patient cost	22,842.86	2348.85
Testing and Medicine cost for isolated cases	6988.71	755.01
Sub-Total	33,304.13	3460.93
Indirect Cost *		
Loss of wages for ICU patients	134.68	13.85
Loss of wages for hospitalized non-ICU patients	1010.11	103.87
Loss of wages for isolated cases	5319.94	574.73
Loss of total yearly wages due to premature death	41,764.88	3483.21
Loss of wages for exposed contacts/individuals	134,681.95	13,848.84
Sub-Total	182,911.56	18,024.50
Govt. of Maharashtra’s fund allocation/expenditure for COVID-19 pandemic *	15,470.03	
Total *	231,685.72	36,955.45
Value of averted cases *	194,730.27	
	(2.60 billion USD) ^#^	
Cost Savings	84.05%	

Note: Infection case figures are rounded to the nearest integer for projected case scenario. * All monetary values are stated in millions of Indian Rupees. ^#^ 1 USD = 75 INR.

## Data Availability

All the data used for analysis and related programming codes are available at https://github.com/preshitambade/COVIDMaharashtra.git (accessed on 20 December 2022).

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
