# Peer review of "COVID-19 Pandemic: Did Strict Mobility Restrictions Save Lives and Healthcare Costs in Maharashtra, India?"

_healthcare, 2023, doi:10.3390/healthcare11142112_

Round 1

Reviewer 1 Report

The paper proposed for my review is entitled "COVID-19 Pandemic: Did harsh mobility restrictions save lives and cost in Maharashtra, India?"

The authors seem to want to measure the health and economic effectiveness of partial or complete lockdowns decided during the Covid19 crisis by the political leaders. The subject is interesting and the field of investigation (Maharashtra region) is conducive to a contributory study. However, according to me, in order to make real contributions, the following points need to be improved:

1. The authors state that the effectiveness of quarantine has been thoroughly measured but not that of lockdowns. This statement is rather inaccurate as a lot of work has been done on the effectiveness of lockdowns (see indicative references). This means that the authors should first identify the elements of effectiveness that have already been demonstrated in the literature in order to formulate their research question, i.e. what they can contribute with their article.

2. The introduction presents a research gap, which is insufficiently developed, and then presents the particular situation of India. The presentation of the Indian context seems to show that there is a lack of work on the health and socio-economic effectiveness of lockdowns in this country. The authors explain this by the fact that research on the pandemic is heavily based on what is reported from high-income countries but little on low- or middle-income countries. This is partly untrue as some work not cited by the authors has focused on the effectiveness of lockdown in low- and middle-income countries, particularly India (see indicative references). Therefore, the authors propose to analyse the impact of severe mobility restrictions during the first wave of the Covid-19 crisis on the spread of the disease and the resulting economies in the state of Maharashtra - one of the poorest countries in the world. This research objective is only partially convincing, as it overlooks previous studies on the effectiveness of lockdown in low- and middle-income countries, particularly in India. This research objective should be clarified after having taken note of the works proposed as indicative references and other works that may deal with this theme. 

3. The method presented and the data mobilised are more interesting because they allow the potential research objective of this article to be made clearer. It appears that the calculation of the reduction in socio-economic costs generated by the lockdown is central to this article. Indeed, by starting from the analysis of the trajectory of the epidemic in the state studied, the authors seem to be producing new results on the cost benefits obtained thanks to lockdown in poor regions. Work on the economic impact has been carried out in rich countries (see indicative references) but little on low and middle income countries. Therefore, by taking the results of work in high-income countries, can the authors explain how they will contribute to this literature?

The authors calculate the cost benefits by calculating the direct and indirect cost savings associated with the lockdown. The method is interesting, but is it inspired by cost-benefit analysis methods used in high-income countries or is it a method created by the authors? Moreover, the calculation of indirect costs from lost wages should be justified. Wouldn't the loss of value added (and thus GDP) be more justified? On the other hand, is the loss of income or value added of the contact cases total? In the case of this region, were the contact cases all isolated?

4. The results of the study first present the health trajectory of the epidemic in the study area. It would be interesting to indicate whether the evolution of the R0 observed in this region (scenario B) is close to that observed at the national level, on the one hand, or in other low- or middle-income regions, on the other.

5. Nevertheless, in my view, the most important contributions of the article are on the economic impact of the lockdown. it should be formulated in this way in the introduction. Furthermore, Table 2 presents the cost calculation table for the two scenarios. This table is very interesting but, once again, the loss of salaries of the contact cases should be more justified as it represents 2/3 of the difference between the two scenarios. Furthermore, is the INR 15.47 billion in government benefits justified in both scenarios? Indeed, one can imagine in scenario A, the government allocation could have been higher.

6. The discussion does not seem to me to be sufficiently focused on the contributions of the article. Indeed, the authors outline the limitations of their study but make little mention of its contributions. The authors state that this is the first study to assess the financial implications of mobility restrictions based on real-time data for the state of MH and for India.  Would it not be possible to highlight more clearly how these results improve our knowledge of the economic efficiency of lockdowns in general or in low- and middle-income areas in particular? It would also be important to mention how the medical results obtained in this study are close to or far from the results obtained by previous studies in India (Prinja et al.; Panneer et al; Mandal et al.).

7. In conclusion, the authors simply state that lockdown measures have saved money for the health system. What forms these savings take and how they may be specific to a low-income region either in their calculation or in their mode of occurrence should be clarified?

Finally, I consider this work to be interesting but not sufficiently backed up by the literature on the effectiveness of lockdowns in the face of Covid-19, particularly in low- and middle-income countries. I recommend that you take note of this literature and then formulate a research question that will allow truly new contributions to emerge.

I thank you for giving me the opportunity to read your study and encourage you to improve your work.

Yours sincerely

Indicative references

Alfano, V., & Ercolano, S. (2021). Stay at home! Governance quality and effectiveness of lockdown. Social indicators research, 1-23.

Amer, F., Hammoud, S., Farran, B., Boncz, I., & Endrei, D. (2021). Assessment of countries' preparedness and lockdown effectiveness in fighting COVID-19. Disaster Medicine and Public Health Preparedness, 15(2), e15-e22.

Galiani, S. (2022). Pandemic economics. Journal of Economic Behavior & Organization, 193, 269.

Homburg, S. (2020). Effectiveness of corona lockdowns: evidence for a number of countries. The Economists' Voice, 17(1).

Li, Y., Undurraga, E. A., & Zubizarreta, J. R. (2022). Effectiveness of localized lockdowns in the COVID-19 pandemic. American Journal of Epidemiology, 191(5), 812-824.

Mandal, S., Das, H., Deo, S., & Arinaminpathy, N. (2020). When to relax a lockdown? A modelling-based study of testing-led strategies coupled with sero-surveillance against SARS-CoV-2 infection in India. medRxiv, 2020-05.

Panneer, S., Kantamaneni, K., Akkayasamy, V. S., Susairaj, A. X., Panda, P. K., Acharya, S. S., ... & Pushparaj, R. R. B. (2022). The great lockdown in the wake of COVID-19 and its implications: lessons for low and middle-income countries. International journal of environmental research and public health, 19(1), 610.

Prinja, S., Bahuguna, P., Chugh, Y., Vassall, A., Pandey, A., Aggarwal, S., & Arora, N. K. (2020). A model based analysis for COVID-19 pandemic in India: implications for health systems and policy for low-and middle-income countries. medRxiv, 2020-06.

Scherbina, A. (2021). Assessing the Optimality of a COVID Lockdown in the United States. Economics of disasters and climate change, 5(2), 177-201.

Sheikh, A., Sheikh, Z., & Sheikh, A. (2020). Novel approaches to estimate compliance with lockdown measures in the COVID-19 pandemic. Journal of global health, 10(1).

Spiliopoulos, L. (2022). On the effectiveness of COVID-19 restrictions and lockdowns: Pan metron ariston. BMC Public Health, 22(1), 1-15.

Yanovskiy, M., & Socol, Y. (2022). Are Lockdowns Effective in Managing Pandemics? International Journal of Environmental Research and Public Health, 19(15), 9295.

Reviewer 2 Report

REVIEW REPORT FOR THE STUDY “COVID-19 PANDEMIC: DID HARSH MOBILITY RESTRICTIONS SAVE LIVES 2 AND COST IN MAHARASHTRA, INDIA?”

Journal: Healthcare

The paper "COVID-19 Pandemic: Did harsh mobility restrictions save lives and cost in Maharashtra, India?", performs a study on the experience of confinement during the first phase of the Covid-19 pandemic, in the state of Maharashtra, where after initial complete shutdown, the state slowly relaxed restrictions. Authors aim to estimate shutdown’s impact on COVID-19 cases and associated healthcare costs. They fitted log-linear models to estimate growth rate. According to the authors, the rapid response of Maharashtra's public health system delayed the pandemic and prevented new cases and deaths in the first wave of the pandemic, resulting in a net saving cost of 84.05%.

Title and summary. The title and abstract express well the object of study, objectives, and results of the article.

Structure of the article. The contents are well organized and they adhere to the IMRaD structure. It includes a theoretical framework of the research problem but at this point, I suggest the authors incorporate some other bibliographic references that I miss in the text:

Albassam D, Nouh M, Hosoi A. The Effectiveness of Mobility Restrictions on Controlling the Spread of COVID-19 in a Resistant Population. Int J Environ Res Public Health. 2023 Mar 31;20(7):5343. doi: 10.3390/ijerph20075343. PMID: 37047958; PMCID: PMC10094504.

Mahmoudi J, Xiong C. How social distancing, mobility, and preventive policies affect COVID-19 outcomes: Big data-driven evidence from the District of Columbia-Maryland-Virginia (DMV) megaregion. PLoS One. 2022 Feb 17;17(2):e0263820. doi: 10.1371/journal.pone.0263820. PMID: 35176031; PMCID: PMC8853552.

Long E, Patterson S, Maxwell K, Blake C, Bosó Pérez R, Lewis R, McCann M, Riddell J, Skivington K, Wilson-Lowe R, Mitchell KR. COVID-19 pandemic and its impact on social relationships and health. J Epidemiol Community Health. 2022 Feb;76(2):128-132. doi: 10.1136/jech-2021-216690. Epub 2021 Aug 19. PMID: 34413184; PMCID: PMC8380476.

Focusing on the opportunity of the study, it must be said that it is useful work since it covers one of the major problems resulting from the Covid-19 health pandemic.

Materials and methods.

Regarding the material and methods section, the methodology is tailored to the object of study and the objectives and is explained in a transparent manner while it has been validly applied to guarantee the results.

Results.

The results are significant and they are presented in an adequate and understandable way not only through narration but also with self-explained tables and figures that are also well elaborated in terms of presentation. The results justify and relate to the objectives and methods and the results are of sufficient interest.

Discussion.

The discussion appropriately compares the study results with other works, highlighting the main study findings. The 97% of the bibliography cited in the study belongs to the previous five years.

However, I would propose the inclusion of three bibliographic references in the discussion section:

Lally M. A cost-benefit analysis of COVID-19 lockdowns in Australia. Monash Bioeth Rev. 2022 Jun;40(1):62-93. doi: 10.1007/s40592-021-00148-y. Epub 2022 Jan 28. PMID: 35088370; PMCID: PMC8794621.

Strzelecki, A.; Azevedo, A.; Rizun, M.; Rutecka, P.; Zagała, K.; Cicha, K.; Albuquerque, A. Human Mobility Restrictions and COVID-19 Infection Rates: Analysis of Mobility Data and Coronavirus Spread in Poland and Portugal. Int. J. Environ. Res. Public Health 2022, 19, 14455. https://doi.org/10.3390/ijerph192114455

Marcu, Silvia. (2021). Towards Sustainable Mobility? The Influence of the COVID-19 Pandemic on Romanian Mobile Citizens in Spain. Sustainability. 13. 10.3390/su13074023.

Overall, it is an interesting study and should be considered for publication in Healthcare, once the minor revisions proposed have been resolved.

Reviewer 3 Report

Thank you very much for the opportunity to review your manuscript. 

Congratulations to the authors, as the work presented for review is excellent. 

The authors have presented a fascinating and insightful analysis and have conducted an interesting discussion. 

The article is very well structured. 

I have only two suggestions that may make the manuscript more readable. 

The first tip concerns the structure of the article. I would suggest that the part between lines: 96-266 be put into one chapter, "Materials and Method". Currently, the methodological part of the paper is blurred. 

A little tip is to consistently edit the paper and use the past tense to describe what the authors did in the analysis, not what they are doing. 

Thanks again, and congratulations.

Reviewer 4 Report

The authors took up the challenge of estimating the effects of introducing a lockdown in one of the states in India. The results showed the number of cases that were avoided and the costs that were not incurred thanks to the lockdown. The article is interesting even though the conclusions are obvious and in line with expectations.

The article lacks a literature review. Several publications were mentioned, but without substantive discussion. It was mentioned that analogous analyzes were carried out in developed countries, but no results were commented on. Studies conducted in both developed and developing countries have not been properly discussed. So the article is more of a report than a scientific analysis.

The positive aspect is that an attempt has been made to calculate both direct and indirect costs. It should be emphasized that the task set by the authors was difficult due to problems with medical and cost data. Therefore, the results are estimates themselves, not calculation results, and should be treated appropriately. The undoubted value of the article is that it is a voice in the discussion concerning, above all, indirect costs. Of course, formula (3) "defined" indirect costs, but in my opinion the authors should present the definitions of indirect costs used in the literature and comment on them. In this approach, the results obtained are completely incomparable to the results obtained in other analyses. The very problem of measuring indirect costs is a challenge for health economics and it would be worthwhile - especially after the pandemic - to join the attempt to formulate a comprehensive definition and create databases, i.e. medical registers and economic databases.

The research method is described as well as the results obtained and the economic analysis. However, I emphasize that this is a local expertise for the needs of local authorities. The conclusions contain seven lines of description without conclusions supported by numerical results. The article does not add much to the achievements of economics, especially the development of a global pandemic and preparation for potentially future pandemics. After all, economic/scientific analysis is not just a computational task.

Reviewer 5 Report

This research investigates the COVID-19 spread and the effectiveness of the restriction measures. The topic is important and interesting. But the presentation of the research has some flaws and needs improvement. 

1. The geographic background of India needs further explanation. Many readers might not be familiar with India and how Maharashtra is important and worth investigating should be addressed.

2. Some statistics to show the severity of Covid-19 are mentioned in the manuscript. But international comparison could be included to highlight the degree of severity.  The current description, containing only the total incidence and deaths, is dry and not able to arouse readers' interest. Comparisons such as reproduction rate and doubling rate could be mentioned.

3. The phases of lockdown are described in detail in terms of date and time in the introduction. But the meaning and measures of lockdown lack explanations.  There are 4 lockdowns during the study period. These seem to follow the categories of government announcements. Are there any differences between the 4 lockdowns? The projections are based on the 4 lockdown time frame. It is not convincing since there is no sign of a pattern changing over the 4 lockdowns. 

4. Two waves of pandemics were mentioned in the manuscript. How are they classified? 

5. Different outcome measures are employed to describe the seriousness of the pandemics including doubling rate, growth rate, R, costs, etc. There should be a section focusing on the outcome measure description.

6. Several R statistical packages are employed for analyses. Many readers might not be familiar with the purposes and functions of the packages. Especially some of the packages serve as the major projection tools in the study. How scenarios A and B work in the models to predict the cases for comparison with and without restrictions need further explanation. Whether these packages have been employed in the extant publications should be discussed in the sections of literature review or method.

6. The figures have very low resolution and the labels are too small to read.

7. The are typos and writing mistakes, such as in lines 107 and 120. 

Round 2

Reviewer 1 Report

Dear authors,

thank you very much for your efforts to improve your article

You present now two major contributions 

Firstly, an analysis of the impact of severe mobility restrictions during the first wave of the pandemic on disease spread and associated economies in the state of Maharashtra - 'one of India's hotspots reported by COVID-19 during the first wave. Secondly, you demonstrate a simple but useful way of modeling health system costs that can be used to assess health costs. 

1. The second contribution (modeling the costs of healthcare systems) should be discussed more in the discussion section based on the literature on this modeling. 

2. You should recall these two essential contributions in the conclusion of your article. 

These are the only requests for modification that I would suggest.

thank you for your future efforts

Kind regards

Reviewer 4 Report

After reading the revised and expanded version of the article, I have the impression that my comments from the review have been taken into account.
